# Intelligent Model for the Reliability of the Non-Intrusive Continuous Sensors Used for the Detection of Fouling-Layer in Heat Exchanger System

**Daniel Osezua Aikhuele** [1,2,*], **Desmond E. Ighravwe** [2] **and Shahryar Sorooshian** [3]

1    Department of Mechanical Engineering, University of Port Harcourt, East-West Road Port Harcourt, Choba 500272, Nigeria
2    Faculty of Engineering and Built Environment, University of Johannesburg, Johannesburg 2006, South Africa
3    Department of Business Administration, University of Gothenburg, 405 30 Gothenburg, Sweden
*    Correspondence: daniel.aikhuele@uniport.edu.ng

**Abstract:** Faults in this sensor must be detected on time to ensure the functionality of the entire system's architecture and to maintain system balance, which will keep false positive rates low during the system's operational period. False positives reduce diagnostic confidence and necessitate unnecessary and costly mitigation actions, lowering system productivity. It is on this basis that this study proposes a clustering model algorithm (K-mean clustering) to investigate and manage the reliability and performance of the sensors. The results from the implementation of the K-mean clustering method show that the running of the algorithm fits the model correctly, both for the training of the dataset and for the prediction of the cluster in each of the datasets considered. A reasonable grouping was found for the two and three clusters considered, which are represented by the colors (blue, orange, and green). These colors indicate the fault state, non-fault state, normal state, and abnormal state of the non-intrusive continuous sensor. The simulated results show the fault state in the blue region and the non-fault state in the orange region for the two clusters, while the normal state is in the blue region and the abnormal state is in the orange and green regions for the three clusters considered.

**Keywords:** non-intrusive continuous sensors; heat exchanger system; K-mean clustering; reliability and performance

## 1. Introduction

Heat exchangers are one of the important systems used in the process and power generation industries [1]. It is used for transferring heat from one medium to another, where such a medium could be in the form of a gas, a liquid, or a combination of both. The media could be separated in some applications by a solid wall, or it could be completely mixed in other applications; however, they are aimed at improving the energy efficiency of such applications by transferring heat from systems where it is not needed to systems where it will be more useful [2]. "For example, waste heat in the exhaust of an electricity-generating gas turbine can be transferred via a heat exchanger to boil water to drive a steam turbine and generate electricity". The most common type of heat exchanger is a car radiator, where the cooling process of the car engine is optimized to make it more efficient.

Heat exchangers work because heat naturally flows from higher temperatures to lower temperatures. Therefore, if a hot fluid and a cold fluid are separated by a heat-conducting surface, heat can be transferred from the hot fluid to the cold fluid [3]. Several types of heat exchangers have been developed for use in steam power generation plants, chemical processing plants, petroleum refinery plants, buildings, heating and air conditioning systems, natural gas processing, and space heating and refrigeration units [4]. Despite the many benefits of the heat exchanger system application in the processing and power

generation industries, it is often challenged by the formation of unwanted material deposits on the heat transfer surfaces. These deposited materials, which are in the form of scale, algae, suspended solids, and insoluble salts on the internal or external surfaces of heat exchangers [5], are sometimes called "heat exchanger fouling," and they occur during process heating or cooling [6].

Although the fouling of heat exchanger systems has been the subject of so much research in the recent past [7], there are still a lot of contributions and improvements that are needed to fully understand how to handle and manage some of the specific heat exchanger fouling problems. Among these problems is how to address reliability issues (fault diagnosis and prognosis) in the non-intrusive continuous sensors that are used in heat exchanger systems. Non-intrusive continuous sensors are mainly deployed in heat exchanger systems for real-time fouling layer detection [8]. These sensors are used as units for monitoring and controlling the heat exchanger system architecture [9], through which faults in the system and system performances are determined.

Considering the very important role of these sensors in the heat exchanger systems used in power generation companies, there is a need for a comprehensive approach to the reliability and performance management of the sensors used in fault detection in the system's architecture. Moreover, a balance is required to keep false-positive rates low during the operational period of the system. False positives reduce diagnostic confidence and necessitate unnecessary and costly mitigation actions, lowering system productivity. The quality of the health management system for the sensors, therefore, can be concluded to have a direct effect on the performance of the entire system. On this basis, this study seeks to investigate and propose a model for the evaluation of the reliability and performance of the sensors used in heat exchanger systems by using a combination of methodologies.

Diagnostic and prognostic analysis methods are fast-growing engineering science tools that are focused on the prediction and health condition management of engineering systems and their components based on their current and previous status. The main goal of the analysis is to provide an accurate prediction of the reliability [10], availability, performance [11,12], and remaining useful life of the system.

Based on this preposition, they are expected to efficiently monitor and track the health status of a system during operation and to determine its reliability and availability, as well as its performance and remaining useful life. Data from the processes are extracted from relevant features associated with the degradation condition and failure of the system or components [13] using sensors. In heat exchanger systems, non-intrusive continuous sensors have been suggested [8]. Recently, it has been found, however, that this sensor is affected by so many operational and environmental conditions [14].

In order to ensure the functionality of the sensor system, there is a need for a methodology and procedure for monitoring and addressing the different problems and challenges associated with the system. Any fault leading to failure in these sub-systems needs to be efficiently identified and properly isolated using the limited set of sensor signals available [15].

Given the importance of non-intrusive continuous sensors developed for real-time detection of fouling layers in heat exchanger systems, which several companies and heat exchanger plants have recently adopted, it is critical that an intelligent model is in place to monitor, evaluate, and study their reliability, availability, and performance. The development of an intelligent model is not only important for the heat exchanger system but will also help to address the several research reports that concluded that the reliability of sensors in heat exchanger systems is the main barrier to their integration into our everyday lives in the factory [16].

In this paper, an intelligent model (the K-mean clustering model) is proposed for addressing and dealing with reliability-related issues in the non-intrusive continuous sensors used for real-time fouling-layer detection in heat exchanger systems. The contributions of the research project and model are summarized as follows:

- The development of the data-driven model for the evaluation of the reliability and performance of the non-intrusive continuous sensors, which, to the best of my knowledge, has not been used or presented in any other literature for heat exchanger To the best of my knowledge, this is the first study to develop and deploy intelligent algorithms for the detection of faults in the non-intrusive continuous sensors used in heat exchanger systems.
- The study contributes to the heat exchanger literature by implementing a new model algorithm for the reliability management of the sensor used in the heat exchanger systems. This is important since it assists management in designing an efficient maintenance policy.
- The use of the model in addressing physical (real-life) reliability problems is another important contribution.
- To the best of my knowledge, this is the first study to develop and deploy several intelligent algorithms for the evaluation of the non-intrusive continuous sensors used in heat exchanger systems.

## 2. Materials and Methods

In this section, the proposed intelligent data-driven model for addressing and handling the reliability and performance management issues of the non-intrusive continuous sensors used in heat exchanger systems is introduced. First, a mathematical model for the reliability and performance of the non-intrusive continuous sensors is presented; this is followed by the introduction of the intelligent data-driven model algorithms that will be used in the analysis of the non-intrusive continuous sensors data.

*2.1. Mathematical Model for the Reliability and Performance of the Non-Intrusive Continuous Sensors*

To fully understand the mathematical concept of the problem the research intends to solve, the following mathematical expression is presented, which describes the challenges associated with the sensors used in the heat exchanger system architecture. The consideration here is that the sensors are subjected to the following environmental conditions: high heat, wear, and vibration, which is assumed to affect their reliability and functionality.

Let $\beta$ be the set of sensor components that constitute the heat exchanger system architecture such that:

$$\beta = \{\beta_1, \beta_2, \beta_3, \ldots, \beta_n\} \tag{1}$$

$$\beta_1 = \left\{\beta_1^1, \beta_1^2, \beta_1^3, \ldots, \beta_1^z\right\} \tag{2}$$

$$\beta_x = \left\{\beta_{2n}^1, \beta_{2n}^2, \beta_{2n}^3, \ldots, \beta_{2n}^z\right\}, \alpha_x \epsilon \ \alpha \tag{3}$$

where $\beta_1^z$ , $\beta_1^z \epsilon \ \beta_1$ are the $a^{th}$ sensor components under the set of sensor components $\beta_1$ that constitutes the heat exchanger system architecture, and $\beta_n^z$ , $\beta_n^z \epsilon \ \beta_n$ are the $z^{th}$ sensor components under the set of sensor components $\beta_n$ that constitutes the heat exchanger system architecture.

Then the reliability and functionality of the heat exchanger system architecture with the set of sensor components $\beta_1^a$ at the epoch $t_x$, $t_x \ \epsilon \ t$, $t = \{t_1, t_2, \ldots, t_x\}$ can be denoted as $R_1(\beta_1^a, t_x)$. Similarly, the reliability and functionality of heat exchanger system architecture with the set of sensor components $\beta_n^z$ at the epoch $t_x$, $t_x \ \epsilon \ t$, $t = \{t_1, t_2, \ldots, t_x\}$ can also be denoted as $R_1(\beta_n^z, t_x)$.

If the failure rate of the sensor components $\beta_1^a$ and $\beta_n^z$ is a variable and is given as $Y_1(\beta_n^z, \beta_1^a, t_x)$ at the epoch $t_x$, then according to the basic concept of reliability calculations, the mean time to failure of the sensor components is therefore given as follows:

$$M_1(\beta_n^z, \beta_1^a, t_x) = \frac{1}{Y_1(\beta_n^z, \beta_1^a, t_x)} \tag{4}$$

If the threshold value for the failure rate of the sensor components $\beta_1^a$ and $\beta_n^z$ is given as $Y_{thresh}$, the sensor components $\beta_1^a$ and $\beta_n^z$ can therefore be said to have a high optimal reliability and a low optimal reliability when $Y_{thresh}(\beta_n^z, \beta_1^a, t_x) \epsilon \{0, 1\}$. The threshold value for the failure rate of the sensor components $\beta_1^a$ and $\beta_n^z$ at epoch $t_x$ is therefore given as follows:

$$Y_{thresh}(\beta_n^z, \beta_1^a, t_x) = 1 \tag{5}$$

$$Y_{thresh}(\beta_n^z, \beta_1^a, t_x) = 0 \tag{6}$$

for a high optimal reliability and a low optimal reliability of the sensor components, respectively. Similarly, the threshold value for the reliability and functionality of heat exchanger system architecture with the set of sensor components at the epoch $t_x$ is given as $R_{thresh}$, where:

$$R_{thresh} = Y_{thresh}(\beta_n^z, \beta_1^a, t_x) \epsilon \{0, 1\} * t_y \tag{7}$$

where $t_y$ is the overall time.

Equation (7) can be used for the computation of high and low optimal reliability of the sensor and actuator components in the heat exchanger system architecture.

### 2.2. Intelligent Data-Driven Model—Clustering Algorithm

The clustering method, which was adopted in this study, is an unsupervised learning method that takes input features and data and does not require proper labels to predict and evaluate them. It is a data analysis technique for identifying intriguing patterns in data, such as fault patterns and groupings. It provides a quick summary of the data that could be utilized to make inferences. Since the purpose of a clustering task is to find data structures, the clustering method must therefore be able to determine the number of structures/groups in the data and how the features are distributed within each group.

Clustering, for example, can be used to detect defects, faults, and anomalies in a system by using the system's database or historical data; moreover, the locations of the faults or defects in the system, as well as the area where errors occur more frequently, can also be determined using the clustering method. There are several clustering methods that can be used for this task, including K-mean clustering [17], mini-batch K-means clustering [18], spectral clustering, Gaussian mixture clustering [17], birch clustering [19], density-based clustering [20,21], hierarchical clustering, and random forest clustering [22]. All of these methods can be successfully implemented to address the reliability problems in the sensor. In this thesis, however, the K-mean clustering model algorithm was proposed, and the other methods are used for the comparison of the final simulation results.

### 2.3. K-Mean Clustering

K-means clustering is a vector quantization approach that seeks to partition $n$ observations into $k$ clusters, with each observation belonging to the cluster with the closest mean (cluster centers or cluster centroid), which serves as the cluster's prototype such that the data space is divided into Voronoi cells as a result of this [23]. Within-cluster variances (squared Euclidean distances) are minimized by K-means clustering but not the regular Euclidean distances. The mean optimizes squared errors, while only the geometric median minimizes the Euclidean distances [24]. The use of K-medians and K-medoids, for example, can lead to better Euclidean solutions.

There are three main characteristics of k-means that make it very efficient for solving engineering problems; however, these same characteristics are also frequently seen as its most significant drawbacks. These characteristics include the following [25]:

1.  Euclidean distance, which is used as both metric and variance for measuring the cluster scatter;
2.  The number of clusters k when used as an input parameter; selecting an incorrect value for k may result in bad results. It is important, therefore, to check the number of clusters in the dataset when performing diagnostic checks with the k-mean clustering method;

3. Finally, the convergence to a local minimum can have an unexpected ("wrong") result.

Although the problem is computationally challenging, effective heuristic techniques quickly converge to a local optimum. Both K-means and Gaussian mixture modeling use an iterative refining method that is comparable to the expectation–maximization algorithm for mixtures of Gaussian distributions. They both use cluster centers to represent the data; however, K-means clustering finds clusters with similar spatial extents, whereas the Gaussian mixture model enables clusters to have diverse shapes.

**Definition 1.** *If a set of observations is given by* $(x_1,\ x_2,\ x_3, \ldots, x_n)$, *where each of the observations is a d-dimensional real vector, the K-means clustering, therefore, aims to partition the n observations into* $k(k \leq n)$ *sets* $S = (S_1,\ S_2,\ S_3, \ldots, S_k)$, *such that the within-cluster sum of squares (WCSS) is minimized as much as possible (i.e., variance). The objective, therefore, is given as follows:*

$$arg_S min \sum_{i=1}^{k} \sum_{X \in S_i} \parallel x - \mu_i \parallel^2 = arg_S min \sum_{i=1}^{k} |S_i| Var\ S_i \tag{8}$$

*where* $\mu_i$ *is the mean of points in* $S_i$, *and it is equivalent to the minimization of the pairwise squared deviations of the different points within the same clusters.*

$$arg_S min \sum_{i=1}^{k} \frac{1}{|S_i|} \sum_{x,y \in S_i} \parallel x - \mu_i \parallel^2 \tag{9}$$

The overall variance is constant, and this is equivalent to maximizing the sum of squared deviations between points in various clusters, and it is equal to the between-cluster sum of squares (BCSS). The algorithm for the implementation of the K-mean clustering method that has been proposed for the reliability and performance management of the non-intrusive continuous sensor was developed using the Python 3 programming language, and it is executed using an online python platform.

## 3. Results

In this section, the non-instructive continuous sensors used in heat exchanger systems, which are expected to efficiently monitor and track the health status of the systems during operation, were investigated to determine their reliability and performance when faced with harsh operational and environmental conditions. This sensor, which is known to be very sensitive, is investigated using the clustering methods presented in the previous chapter. Here, the reliability and performance of the sensor are measured by attempting to use the clustering model algorithms to detect faults and anomalies in the huge data they normally generate. In the evaluation of the reliability and performance of the non-instructive continuous sensor, a case study approach was adopted, as presented in the section below.

### 3.1. Case Study of a Non-Instructive Continuous Sensor in a Heat Exchanger System

In this paper, a non-instructive continuous sensor used for monitoring and dealing with fouling problems in a heat exchanger system in a food manufacturing company was investigated. The non-instructive continuous sensor, which itself has generated much data (big data) over the past two years as it relates to the operations of the heat exchanger system and its components, is believed to conceal knowledge about the sensor, hence its investigation. First, the following failure modes of the sensors, as shown in Table 1, are considered.

**Table 1.** Failure modes, causes, and effects in the non-intrusive continuous sensor.

| Failure Modes | Failure Cause | Failure Effects |
|---|---|---|
| No output | Faulty component | No immediate effect, replacement required |
| | Short-circuiting and open circuiting | |
| High output | Unknown | No immediate effect, undetected failure |
| Low output | Unknown | Excessive oil temperature to TCV; possible gas turbine emergency shut-down |
| Latch-up | Positive or negative voltage spike on an input or output pin of a digital chip that exceeds the rail voltage | |
| Dielectric breakdown | Failure of an insulating material | Flow of current under applied electrical stress. |
| Bridging faults | Connection problems | |

This is particularly important, as the failure modes are also entered as features used in the evaluation process using the clustering model algorithms for sensor reliability and performance. In the first instance, a test dataset of 1000 samples is generated, and two features in some particular period of the sensor's operation are simulated, in which three states, that is, the normal state, the abnormal state, and the fault state of the sensor, are identified. However, this is obtained by splitting the data into two or three clusters, respectively. The results generated from the two clusters are used to determine the fault and non-fault states of the sensor. Similarly, the three clusters are used to determine the normal state of the sensor and the abnormal state. It is worth noting that all of the experiments were carried out on the same hardware and software platforms, with the following specifications: Intel(R) Core (TM) i5-3320M CPU @ 2.60 GHz, 4.00 GB RAM.

### 3.2. Implementation of the K-Mean Clustering Algorithm

Upon the implementation of the K-mean clustering algorithm, the simulated results from the algorithm are presented in Figures 1 and 2 below. The results with respect to the two and three clusters used, respectively, in this study, show the two states for the non-intrusive continuous sensor, that is, the fault state (orange region) and non-fault state (blue region) for the two clusters, and the normal state (blue region) and the abnormal state (orange and green region) for the three clusters, respectively. These prepositions, however, are based on the fault mode rules as presented in [26].

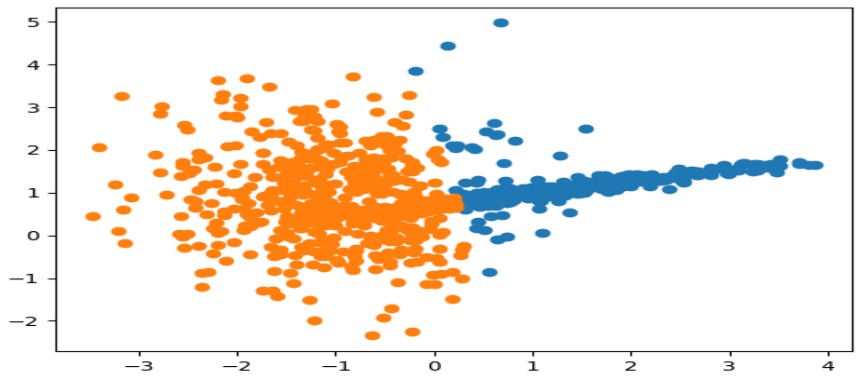

**Figure 1.** Simulated results for the two clusters showing two states of the sensor with 2 features.

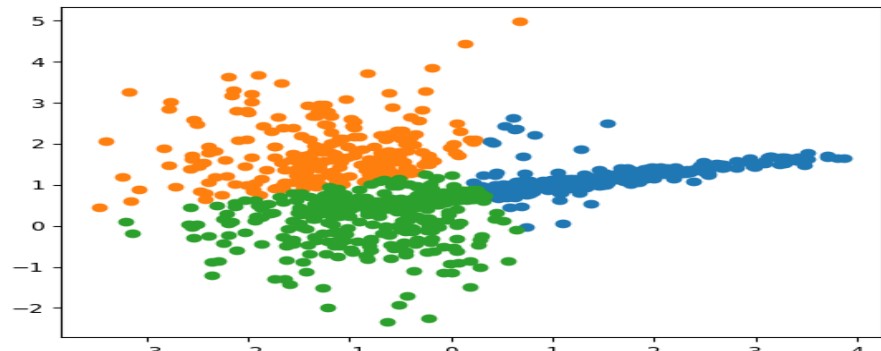

**Figure 2.** Simulated results for the three clusters showing two states of the sensor with 2 features.

Similarly, the K-mean clustering algorithm was implemented for the six (6) features of the sensor by using the same test dataset of 1000 samples over the same period. The simulated results showing the fault state (blue region) and the non-fault state (orange region) for the two clusters and the normal state (blue region) and the abnormal state (orange and green region) for the three clusters, respectively, are presented in Figures 3 and 4, respectively.

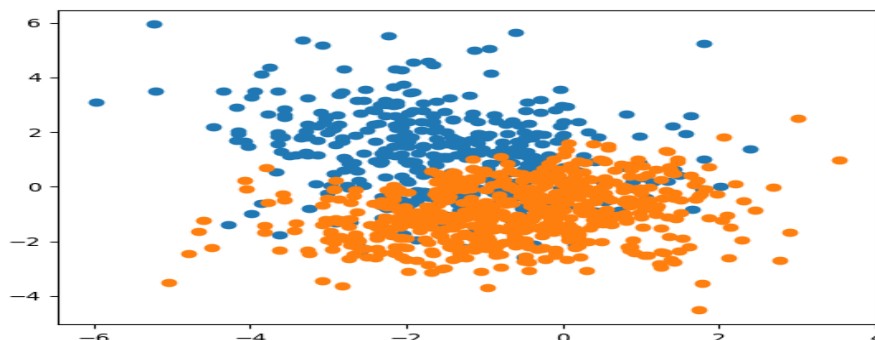

**Figure 3.** Simulated results for two clusters showing states of the sensor with 6 features.

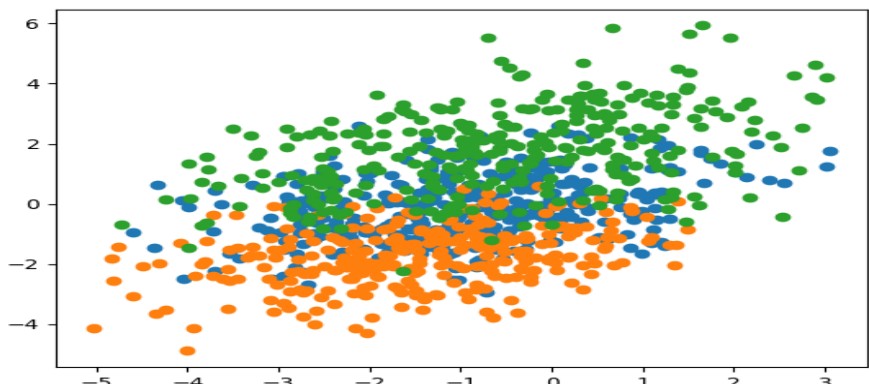

**Figure 4.** Simulated results for three clusters showing states of the sensor with 6 features.

From the above-simulated results, it is easy to see that the running of the algorithm fits the model correctly, both for the training of the dataset and for the prediction of the cluster in each of the datasets considered. This was simulated in the results for the two and six-feature datasets, respectively. For the two and three clusters, a reasonable grouping was found, which was represented by the colors blue, orange, and green. These colors indicate the fault state, non-fault state, normal state, and abnormal state of the non-intrusive continuous sensor. The simulated results generate a scatter plot of the dataset, separated

into different colors in the form of a cluster. The clusters are generated by the association of the different datasets (observations) using the nearest mean. It can be concluded, therefore, that with the implementation of the K-mean clustering algorithm for the analysis of data generated from the non-instructive continuous sensor used in the heat exchanger system, the reliability and performance of the sensor can be managed effectively. Since the fault state, non-fault state, normal state, and abnormal state of the sensors, as well as that of the heat exchanger system, can be easily identified.

### 3.3. Comparison of the Simulation Results with Similar Clustering Model Algorithms

As stated in Section 3.2, there are several clustering methods that can be used for the management of the reliability and performance of the non-intrusive continuous sensor; among them are K-mean clustering, mini-batch K-means clustering, spectral clustering, Gaussian mixture clustering, birch clustering, density-based clustering, hierarchical clustering, and random forest clustering. All of these methods, however, can be successfully implemented to address the reliability and performance evaluation problems. In comparing the simulated results presented when the K-mean clustering model algorithm was implemented, the following model algorithms were applied in this paper to compare the simulated results: the mini-batch K-means clustering model, the spectral clustering model, the Gaussian mixture clustering model, and the birch clustering model algorithms. The simulation is implemented in the online Python 3 platform using the same dataset of 1000 samples obtained from the failure modes (6 features) of the non-intrusive continuous sensors. The simulated results for the different clustering model algorithms when the dataset is grouped into two clusters are presented in Figure 5.

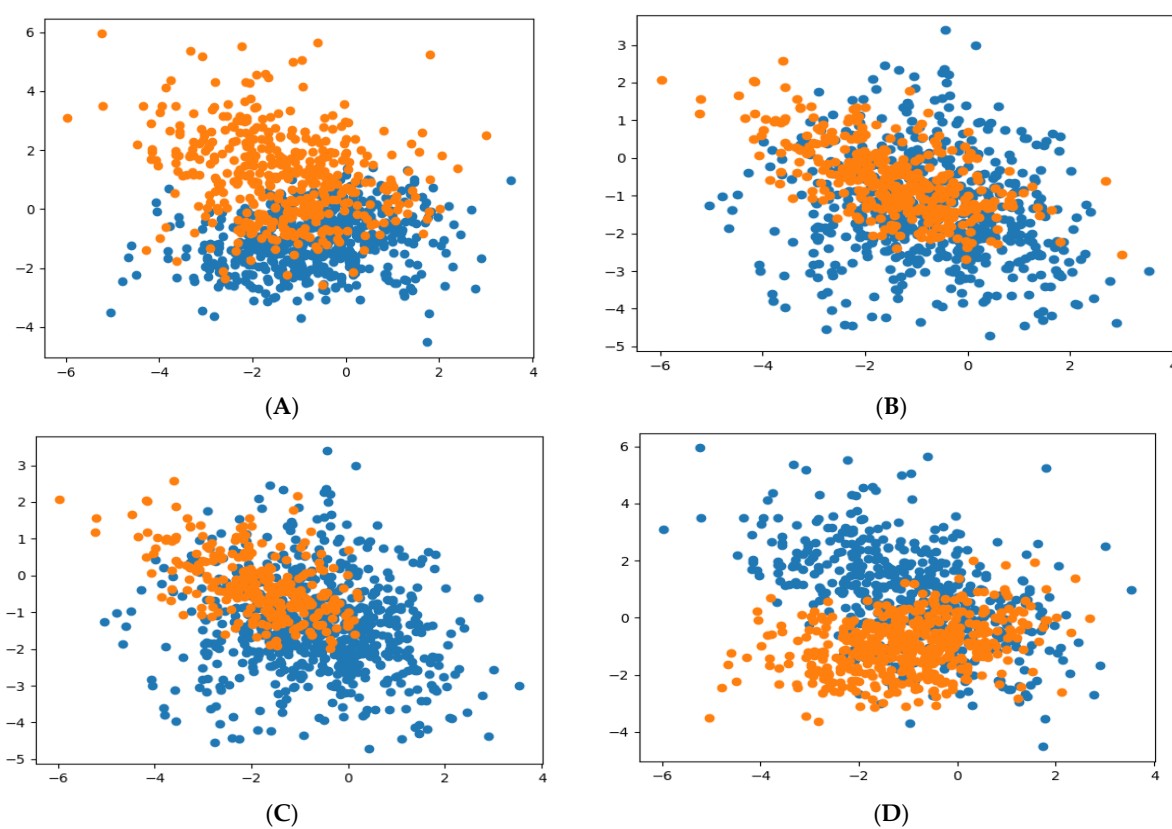

**Figure 5.** Simulated results for two clusters showing states of the sensor for the algorithms: mini-batch K-means clustering (**A**), spectral clustering (**B**), Gaussian mixture clustering (**C**), and birch clustering (**D**).

From the simulated results (Figure 5), which compare the different clustering model algorithms, it is not difficult to see that the running of the algorithm fits the model correctly, both for the training of the dataset and for the prediction of the cluster in each of the datasets considered. This was virtualized in the results for the six features dataset, which was drawn from the failure modes in the non-intrusive continuous sensor in the heat exchanger system.

For the two clusters, a reasonable grouping was found, and it is represented by the colors blue and orange. These colors indicate the fault state (orange region) and non-fault state (blue region) for the mini-batch K-means clustering (A); the fault state (blue region) and non-fault state (orange region) for the spectral clustering (B); the fault state (blue region) and non-fault state (orange region) for the Gaussian mixture clustering (C); and finally, the fault state (blue region) and non-fault state (orange region) for the Birch clustering (D). The simulated results generate scatter and dense plots of the dataset, which are separated and identified by the different colors in the cluster. The clusters are generated by the association of the different datasets (observations) using the nearest mean. In Table 2, the simulated results for the clustering model algorithms that were implemented for managing the reliability of the non-intrusive continuous sensor are compared. The results show consistency with the K-means clustering model algorithm, the spectral clustering model algorithm, the Gaussian mixture clustering model algorithm, and the birch clustering model algorithm.

**Table 2.** Comparison of the simulated results for the different clustering model algorithms when two clusters are considered.

| S/N | Clustering Model Algorithms | Faults State | Non-Faults State |
|-----|----------------------------|--------------|------------------|
| 1 | K-means clustering | Blue region | Orange region |
| 2 | Mini-batch K-means clustering | Orange region | Blue region |
| 3 | Spectral clustering | Blue region | Orange region |
| 4 | Gaussian mixture clustering | Blue region | Orange region |
| 5 | Birch clustering | Blue region | Orange region |

In comparing the simulated results presented when the K-mean clustering model algorithm was implemented with three clusters, the following clustering model algorithms were compared: the mini-batch K-means clustering model, the spectral clustering model, the Gaussian mixture clustering model, and the birch clustering model algorithms. The simulation was implemented in the online Python 3 platform using the same dataset of 1000 samples obtained from the failure modes (six features) of the non-intrusive continuous sensors. The simulated results for the different clustering model algorithms when the dataset is grouped into three clusters are presented in Figure 6.

From the simulated results (Figure 6), which compare the different clustering model algorithms, it is not difficult to see that the running of the algorithm fits the model correctly, both for the training of the dataset and for the prediction of the cluster in each of the datasets considered. This was virtualized in the results for the six features dataset, which was drawn from the failure modes in the non-intrusive continuous sensor in the heat exchanger system.

For the three clusters, a reasonable grouping was found, and it is represented by the colors blue, green, and orange. These colors indicate the normal state (orange region) and the abnormal state (green and blue region) for the mini-batch K-means clustering (A); the normal state (blue region) and the abnormal state (orange and green region) for the spectral clustering (B); the normal state (blue region) and the abnormal state (orange and green region) for the Gaussian mixture clustering (C); and finally, the normal state (green region) and the abnormal state (orange and blue region) for the birch clustering (D). The simulated results generate scatter and dense plots of the dataset, which are separated and identified by the different colors in the cluster. The clusters are generated by the association of the different datasets (observations) using the nearest mean. In Table 3, the simulated results

for the clustering model algorithms that were implemented for managing the reliability of the non-intrusive continuous sensor are compared. The results show consistency with the K-Means clustering model algorithm, the spectral clustering model algorithm, and the Gaussian mixture clustering model, respectively.

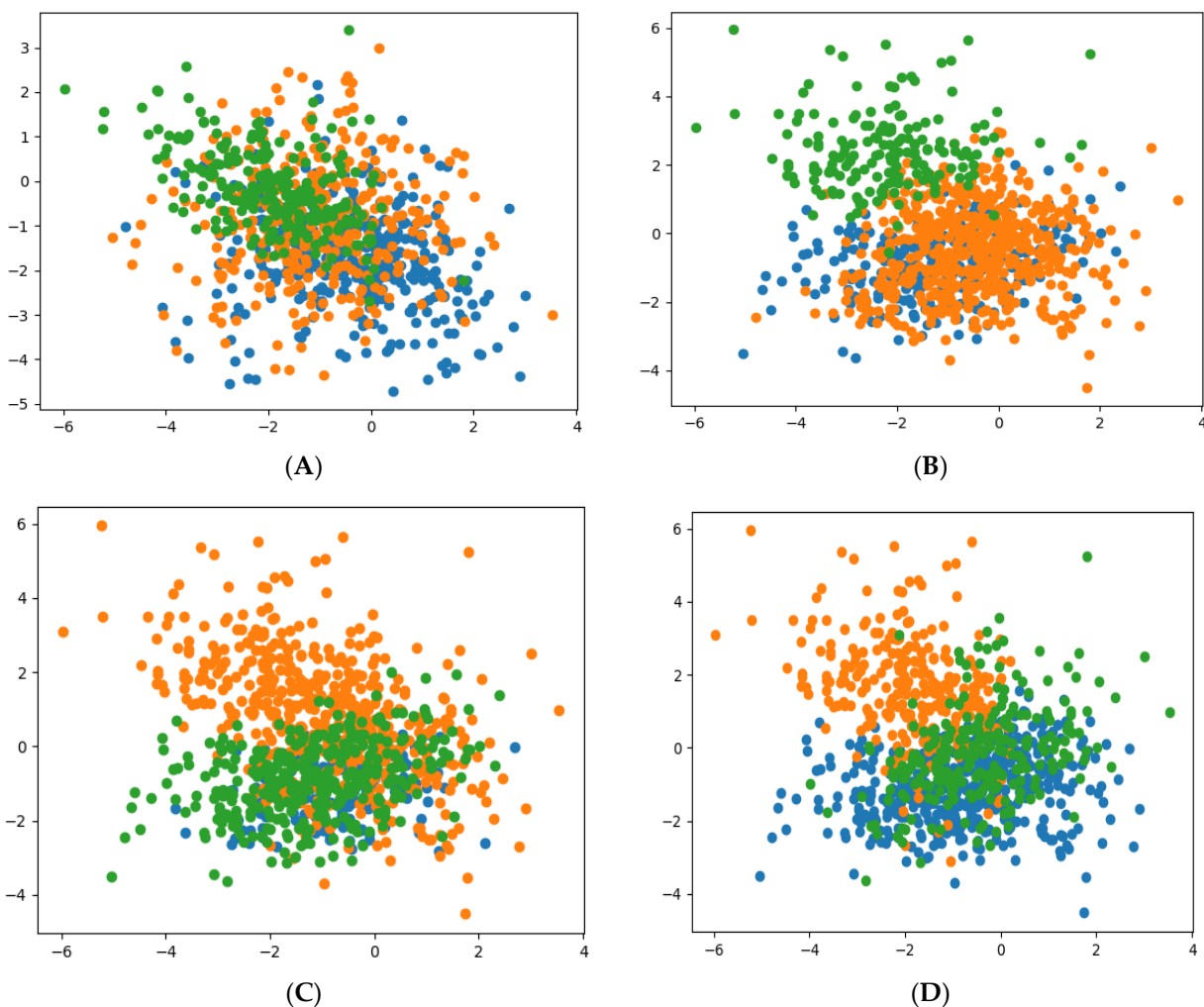

**Figure 6.** Simulated results for three clusters showing states of the sensor for the algorithms: mini-batch K-means clustering (**A**), spectral clustering (**B**), Gaussian mixture clustering (**C**), and birch clustering (**D**).

**Table 3.** Comparison of the simulated results for the different clustering model algorithms when three clusters are considered.

| S/N | Clustering Model Algorithms | Normal State | Abnormal State |
|-----|-----------------------------|--------------|----------------|
| 1 | K-means clustering | Blue region | Orange and green region |
| 2 | Mini-batch K-means clustering | Orange region | Blue and green region |
| 3 | Spectral clustering | Blue region | Green and orange region |
| 4 | Gaussian mixture clustering | Blue region | Green and orange region |
| 5 | Birch clustering | Green region | Blue and orange region |

The following conclusions can be drawn from the compared simulated results for the non-intrusive continuous sensor used in heat exchanger systems that were analyzed using the different clustering model algorithms presented above:

- When the distribution of the dataset is unknown or unclear, it cannot achieve a perfect clustering result, even when two or three clusters are used for the analysis.
- A single clustering mean cannot yield a perfect clustering result.
- In clustering applications, traversing the samples several times is unavoidable.

## 4. Conclusions

In order to manage the reliability and performance of the non-intrusive continuous sensor, a clustering model algorithm was proposed and implemented using the historical failure mode data from the sensor. The K-mean clustering model algorithm proposed in this paper was validated using the following models: the mini-batch K-means clustering model, the spectral clustering model, the Gaussian mixture clustering model, and the birch clustering model algorithm. The study can therefore conclude that the clustering method proposed (the K-mean clustering model algorithm) is an excellent model.

Although the research study's goal and objectives were met and successfully presented in the thesis, there are still many suggestions and questions that need to be addressed in future research work, including:

- What is the algorithm's performance on each of the features and datasets? The algorithm's performance for data other than the data used in this investigation could be drastically different. It would be fascinating to examine how alternative data, such as data from other fault modes in the sensor and the heat exchanger system, compared to those utilized in this study.
- The determination of a faster and more reliable method for estimating the number of clusters used in the implementation of the model algorithms.
- The need to design a self-learning diagnosis system that employs the proposed technique updates the model for each new failure and re-estimates the number of clusters as new data are accumulated. It would be interesting to see if the previous model and earlier estimates of the number of clusters might be used to speed up the calculation and maybe increase performance when updating the model as new data are added.

**Author Contributions:** Conceptualization, D.O.A.; methodology, D.O.A.; software, D.O.A.; validation, D.O.A., D.E.I. and S.S.; formal analysis, D.O.A.; investigation, D.O.A.; resources, D.O.A.; data curation, D.E.I.; writing—original draft preparation, D.O.A.; writing—review and editing, D.O.A.; visualization, D.O.A.; supervision, D.E.I. and S.S.; project administration, D.O.A.; funding acquisition, S.S. and D.E.I. All authors have read and agreed to the published version of the manuscript.

**Funding:** This research received no external funding.

**Institutional Review Board Statement:** Not applicable.

**Informed Consent Statement:** Not applicable.

**Data Availability Statement:** Data are not available to the public.

**Conflicts of Interest:** The authors declare no conflict of interest.

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
