# Peer review of "Intelligent Model for the Reliability of the Non-Intrusive Continuous Sensors Used for the Detection of Fouling-Layer in Heat Exchanger System"

_applsci, doi:10.3390/app13053028_

Round 1
Reviewer 1 Report
The manuscript is written well by the authors and the study is interesting. However, there are some queries that must be carefully addressed by the authors before the manuscript could be accepted for publication in this journal of high repute.
Some of the suggestions/queries are as follows:
1. Language editing is required.
2. What limitations have been observed so far, and how the performance/reliability of sensors are affected by the operational and environmental conditions?
3. No information is provided on the types of clustering models.
4. In the introduction section, kindly provide the (non-mathematical) methodology to be used by the authors, in brief.
5. In line 139, why "if" is used?
6. In line 151, "...components we, therefore...", is we required here?
7. As indicated in eq. 4, will the mean time to failure of the sensor components be simply reciprocal of the failure rate of the sensor components? Is it that simple to be expressed?
Author Response
The manuscript is written well by the authors and the study is interesting. However, there are some queries that must be carefully addressed by the authors before the manuscript could be accepted for publication in this journal of high repute.
Some of the suggestions/queries are as follows:
- Language editing is required.
Thanks so much the manuscript have been edited
- What limitations have been observed so far, and how the performance/reliability of sensors are affected by the operational and environmental conditions?
Thanks so much the limitations have been presented in the form of future research work from the study.
- No information is provided on the types of clustering models.
Thanks so much. Several types of clustering models where mention (see section 2.2)
- In the introduction section, kindly provide the (non-mathematical) methodology to be used by the authors, in brief.
Thanks so much. The non-mathematical methodology used by the authors are discussed in section 2.3.
- In line 139, why "if" is used?
Thanks so much, this has been corrected.
- In line 151, "...components we, therefore...", is we required here?
Thanks so much, this has been corrected.
- As indicated in eq. 4, will the mean time to failure of the sensor components be simply reciprocal of the failure rate of the sensor components? Is it that simple to be expressed?
Thanks so much. Yes is corrected. Mean time to failure is the reciprocal of the failure rate.

Reviewer 2 Report
This paper presents a clustering model algorithm (K-Mean clustering) to investigate and manage the reliability and performance of sensors used for real-time detection in heat exchanger systems. However, there are plenty of things that need to be clarified or improved before it can be considered for possible publication:
1. Moderate editing of English language and style is highly recommended. Here are some examples for further reference:
(1) In page 1 line 25, ‘The simulated results shows the fault state…’, ‘shows’ should be replaced by ‘show’.
(2) In page 2 line 85, ‘Recently it have been found however…’, ‘have’ should be replaced by ‘has’.
(3) In page 5 line 223, ‘The algorithm for the implementation of the K-mean clustering method that have been proposed…’, ‘have’ should be replaced by ‘has’.
(4) In page 9 line 332, ‘The results shows consistency with….’, ‘shows’ should be replaced by ‘show’.
There are still many grammar mistakes identified by the reviewer. However, it is impossible to list all of them here. The author should spend more time on grammar check and proofreading.
2. In Figures 1-6, a legend should be added to each individual figure, which can help readers understand what different colours stand for.
3. In page 5, lines 220 - 225, the authors state that the algorithm is developed using Python 3 programming and executed using an online Python platform. It is recommended that more details about the development platform should be introduced.
4. In Section 3.3, the comparison of simulation results with different clustering model algorithms is insufficient. More in-depth analysis and discussions are required.
5. In the Conclusion part, the authors should state the most important outcome of the current work. It is unnecessary to repeat the background/opinions/methods already presented in the main body. Instead, the authors should interpret and summarize their findings and conclusions at a higher level of abstraction. Therefore, the Conclusion part should be revised carefully.
Author Response
This paper presents a clustering model algorithm (K-Mean clustering) to investigate and manage the reliability and performance of sensors used for real-time detection in heat exchanger systems. However, there are plenty of things that need to be clarified or improved before it can be considered for possible publication:
- Moderate editing of English language and style is highly recommended. Here are some examples for further reference:
Thanks so much the manuscript have been edited
- In page 1 line 25, ‘The simulated results shows the fault state…’, ‘shows’ should be replaced by ‘show’.
Thanks so much, this has been corrected
- In page 2 line 85, ‘Recently it have been found however…’, ‘have’ should be replaced by ‘has’.
Thanks so much, this has been corrected
- In page 5 line 223, ‘The algorithm for the implementation of the K-mean clustering method that have been proposed…’, ‘have’ should be replaced by ‘has’.
Thanks so much, this has been corrected
- In page 9 line 332, ‘The results shows consistency with….’, ‘shows’ should be replaced by ‘show’. There are still many grammar mistakes identified by the reviewer. However, it is impossible to list all of them here. The author should spend more time on grammar check and proofreading.
Thanks so much, these mistakes has been corrected and the manuscript proofread.
- In Figures 1-6, a legend should be added to each individual figure, which can help readers understand what different colours stand for.
Thanks so much. These colours are well discussed and presented in Table 2 and 3 for each of the cases studied.
- In page 5, lines 220 - 225, the authors state that the algorithm is developed using Python 3 programming and executed using an online Python platform. It is recommended that more details about the development platform should be introduced.
Thanks so much. This has been edited in the manuscript.
- In Section 3.3, the comparison of simulation results with different clustering model algorithms is insufficient. More in-depth analysis and discussions are required.
Thanks so much. The discussion presented in the manuscript are just within the scope of the study, however more work has been recommended for future study.
- In the Conclusion part, the authors should state the most important outcome of the current work. It is unnecessary to repeat the background/opinions/methods already presented in the main body. Instead, the authors should interpret and summarize their findings and conclusions at a higher level of abstraction. Therefore, the Conclusion part should be revised carefully.
Thanks so much, this has been corrected

Reviewer 3 Report
This paper applied the clustering method for the detection of fouling layer in heat exchange system. The results and method proposed are useful for applications in heat exchange systems.
However, the methodology introduction is rather rough. More details regarding different clustering methods need to be provided.
The writing of the introduction section needs to be improved. For example, page 1 line 41: "where the cooling process of the car engine is optimize to make it more efficient" should be "optimized".
The figures should include axis names and figure legends.
Author Response
This paper applied the clustering method for the detection of fouling layer in heat exchange system. The results and method proposed are useful for applications in heat exchange systems.
- However, the methodology introduction is rather rough. More details regarding different clustering methods need to be provided.
Thanks so much. Several types of clustering models where mention (see section 2.2)
- The writing of the introduction section needs to be improved. For example, page 1 line 41: "where the cooling process of the car engine is optimize to make it more efficient" should be "optimized".
Thanks so much. This has been edited in the manuscript.
- The figures should include axis names and figure legends.
Thanks so much. Details of every of the Figures are discussed after they have been presented in the manuscript. Hence there will not been need for such repetitions.

Round 2
Reviewer 2 Report
The authors have managed to addressed the reviewer's comments.
Reviewer 3 Report
The comments have been addressed.